# Tooth wear as an indicator of acculturation process in remote Amazonian populations

David Normando[1], Mayara Silva Barbosa[2], Paulo Mecenas[3], Cátia Quintão[2]*

1 Dental School, Department of Orthodontics, Federal University of Pará, Belém, Pará, Brazil, 2 Dental School, Department of Orthodontics, State University of Rio de Janeiro-UERJ, Rio de Janeiro, Brazil, 3 Post-graduation Program of Dentistry, Federal University of Pará-UFPA, Belém, Pará, Brazil

* catiacaq@gmail.com

## Abstract

Riverine populations are typical of the Amazon region that depend on nature for subsistence. These people are considered an intermediate population between the urban and indigenous, the original Amazon habitants. The aim of this cross-sectional study was to evaluate the relationship between tooth wear and age in a remote riverine population from the Amazon, located by the Tucumanduba River (n = 94), and to compare them to previous findings obtained from semi-isolated indigenous (n = 223) and urban populations (n = 40) from the Amazon region, which were examined using the same methodology. Using linear regression, tooth wear explained 54.5% of the variation in the ages of the riverine subjects (p<0.001). This coefficient is mid-way between those obtained in semi-isolated indigenous populations (65–86%) and urban subjects (12%) living in the Amazon. Our findings suggest that tooth wear, a direct evidence of what an individual ate in the past, may be an indicator of the acculturation process in remote populations.

## Introduction

Several methods can be used to estimate chronological age in humans [1–5]. One of the most traditional is the analysis of the degree of mineralization and tooth eruption [1, 6–8]. However, this methodology cannot be used in estimating the chronological age of mature individuals, without a trace of growth, as once at this stage all teeth are already formed and in position. Thus, a measurement parameter that changes continuously over time, without any recovery or neoformation, would be ideal.

Dental wear can be an effective tool for estimating age in different populations [9–11]. As dental hard tissues undergo progressive loss due to mechanical and/or chemical wear and are not replaced [12, 13], quantifying dental wear to estimate chronological age is valid, especially, after development of the dentition and growth cessation. Though, factors such as diet, cultural habits, malocclusion severity and gastroesophageal reflux may modify the degree of wear and its relationship with age according to the investigated population [14, 15].

In the Amazon live the riverine people, a population that depends on the river network for their subsistence and transportation. The local economy consists of small-scale fishing and

**Data Availability Statement:** All relevant data are within the manuscript and its figures and supporting Information files.

**Funding:** The authors received no specific funding for this work.

**Competing interests:** The authors have declared that no competing interests exist.

agriculture. The food is based on the consumption of regional roots, cassava flour, fruits, vegetables and meat from wild animals [16]. During daily tasks it is not uncommon for them to use their teeth as tools [17]. The presence of such habits reveals the cultural origin of these individuals that come from indigenous communities.

In a previous study conducted in the Amazon region, a high correlation was observed between dental wear and chronological age in semi-isolated indigenous communities, thus making it possible to predict the age of an individual through dental wear with relative reliability. Meanwhile, the association was low when a sample of an urban Amazonian population was evaluated [11]. Riverine communities are considered to be an intermediate population between indigenous and urban, not living as isolated as the indigenous populations, but, relatively distant from urban centers. Recently, however, with the increasing urbanization of the Amazon and financial assistance programs from the Brazilian government, there has been greater contact of the riverine people with urban centers [16, 18]. This increased the acculturation process, characterized by cultural changes derived from the contact between two distinct cultures, and assimilation and/or loss of traditional customs may occur [19–21].

Therefore, this study aims to investigate the correlation between tooth wear and age in a riverine population of the Amazon and compare with data previously obtained from indigenous and urban populations in the same region. The purpose of this comparison is to understand the effects of the acculturation process, especially the changes in eating habits, in the Amazon populations.

## Materials and methods

### Ethical considerations

This work was approved by the Research Ethics Committee of the Institute of Health Sciences of the Federal University of Pará under the protocol number 2.055.941, dated May 2017, S1 File. For inclusion in this study participants over 18 years or parents and/or guardians responsible for individuals under 18 years signed an informed consent form.

### Study design

This is a cross-sectional study and conforms to the STROBE guidelines. [22]. The group studied belongs to the riverine community that inhabits the sides of the Tucumanduba River, in the municipality of Abaetetuba, Pará state, Amazonia, Brazil. This community lives in relative isolation, without basic sanitation and with limited access to urban centers. The participants' recruitment and data collection occurred from September 2017 to April 2018.

### Participants and eligibility criteria

The sample of this study included ninety-four (94) riverine individuals from the community of Tucumanduba, between 13 and 61 years old, of both sexes. Personal information, including name, age, sex and address were recorded.

To be included in the study, individuals should have permanent dentition with all teeth erupted, excluding third molars, and no more than seven tooth losses, since a high number of missing teeth could influence the amount of dental wear found on the rest of the dentition.

This population was compared to the indigenous and urban populations evaluated in a previously published research article [11], following the same methodology, composed of 223 indigenous individuals from Middle Valley Xingu, Pará, Brazil, and 40 individuals residing in an urban center, Belém, Pará, Brazil.

## Variables analyzed, data sources, and measurements

The variables considered in the sample were the amount of occlusal tooth wear and the participants' chronological ages. To measure occlusal tooth wear, a previously described and widely applied index (the modified Mockers index) was used, [11, 23–25]. Occlusal faces of the second and first premolars and incisal faces of canines, lateral and central incisors in both dental arches were clinically examined, always using good quality lighting. Then scores were recorded for each tooth: 0 = no wear; 1 = enamel wear only; 2 = dentin wear, with the occlusal/incisal face showing more enamel than dentin; 3 = dentin wear, with the occlusal/incisal face showing more dentin than enamel; 4 = advanced wear stage, near or with pulp exposure (Fig 1). The dental examination was performed by one previously calibrated researcher with experience in this measurement. The reliability of the method was tested after evaluating 20% of the sample, to then examine the entire sample. The assessment was performed using natural daylight and a flashlight, with an assistant recording the measures.

Since a previous analysis of the tooth wear regression coefficient of the evaluated teeth (incisors, canines and premolars) found a weak to moderate association with chronological age, therefore, for each individual, an arithmetic mean of tooth wear was calculated, as described by Normando et al. [24, 25]. The birth dates of the participants were recorded according to birth records.

## Statistical analysis

The determination of chronological age through tooth wear was statistically analyzed using simple linear regression using the MINITAB 17 Statistical Software package (Minitab Inc, State College, PA, USA). Intraclass correlation was used to test the reliability of the tooth wear evaluation. A significance level of 5% was used for all the analyses.

## Results

### Participants

Initially, 95 individuals were deemed to be eligible and were examined, however, the birth date of one participant could not be confirmed, so he was not included in the analysis. Thus, the sample analyzed in the Tucumanduba riverine community was composed of 94 individuals, 42 females (44.7%) and 52 males (55.3%), with a mean age of 24.7 years (13.1–61.9). No significant differences were observed between males and females regarding dental wear (p> 0,05), and for this reason the results were combined.

### Outcome data

Means and ranges of tooth wear values for each population are described in Table 1. The intraclass correlation reveals excellent reproducibility for tooth wear measurements (r = 0.78–0.94, p<0.0001).

In the riverine population, a significant association was observed between occlusal tooth wear and chronological age ($R^2$ = 0.55, p< 0.001) (Fig 2). Wear measurement explained 55% of the age variability in these individuals. This coefficient of determination is intermediate between those found in semi-isolated indigenous populations ($R^2$ = 0.86–0.65) and the urban population of the Amazon ($R^2$ = 0.12).

## Discussion

The results of this study revealed a significant association between dental wear and chronological age in a sample of Amazonian riverines. The measurement of wear explained 55% of the

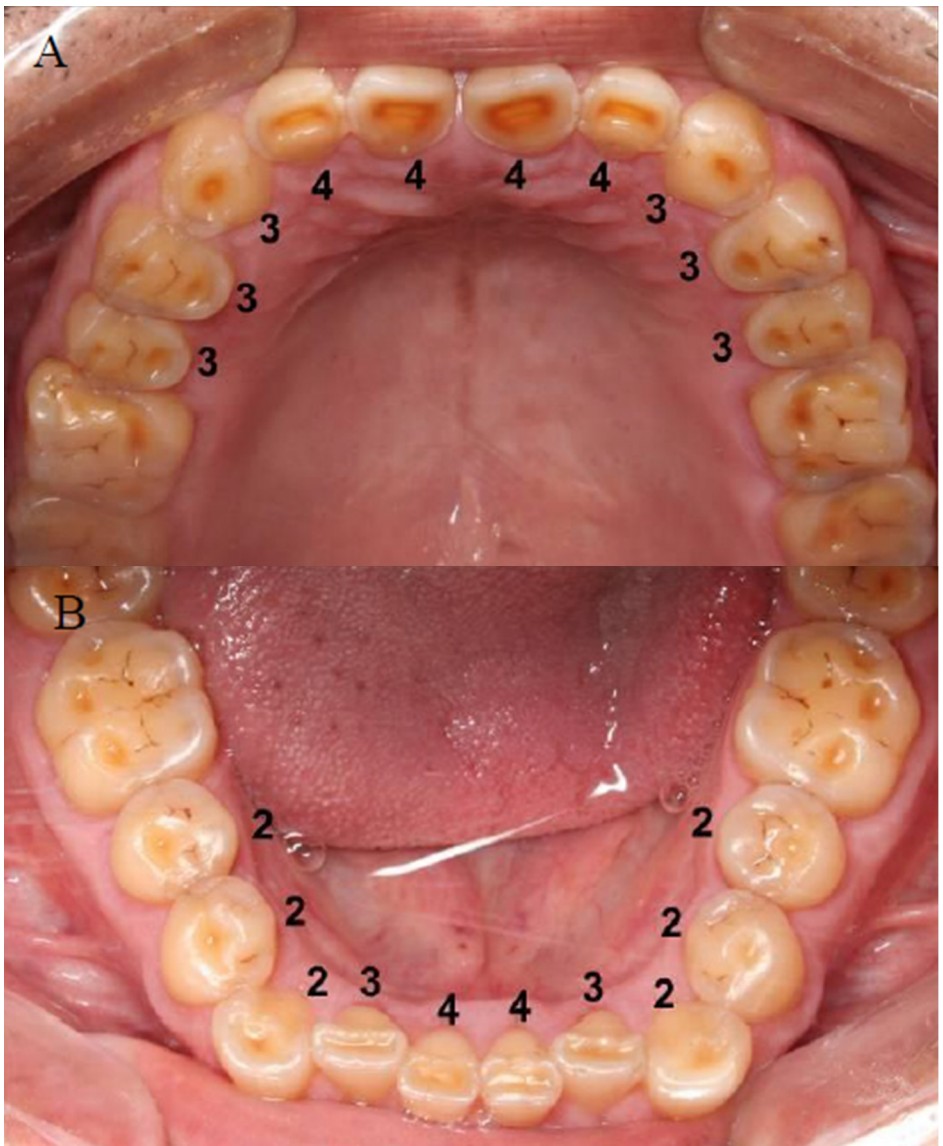

**Fig 1. Modified tooth wear measurement index.** A—upper dental arch; B—lower dental arch; 0—absence of wear; 1—enamel wear only; 2—dentin wear, with the occlusal/incisal face showing more enamel than dentin; 3—dentin wear, with the occlusal/incisal face showing more dentin than enamel; 4—advanced wear stage, near or with pulp exposure.

**Table 1. Sample size (n), mean and range of tooth wear and age of riverine groups, indigenous (Assurini, Xicrin-Kaiapó and Arara) and urban (Belém, Pará).**

| Group | n | Mean of tooth wear (Min-Max) | Age (years) (Min-Max) |
|---|---|---|---|
| Riverine | 94 | 0.64 (0.0–2.6) | 24.65 (13.1–61.9) |
| Assurini | 46 | 0.82 (0.0–2.3) | 19.16 (10.8–45.5) |
| Xicrin-Kaiapó | 60 | 0.63 (0.0–2.3) | 21.51 (10.8–49.3) |
| Arara | 117 | 0.91 (0.0–2.9) | 21.27 (10.3–48.1) |
| Belém | 40 | 0.90 (0.2–1.6) | 22.25 (13.1–42.4) |

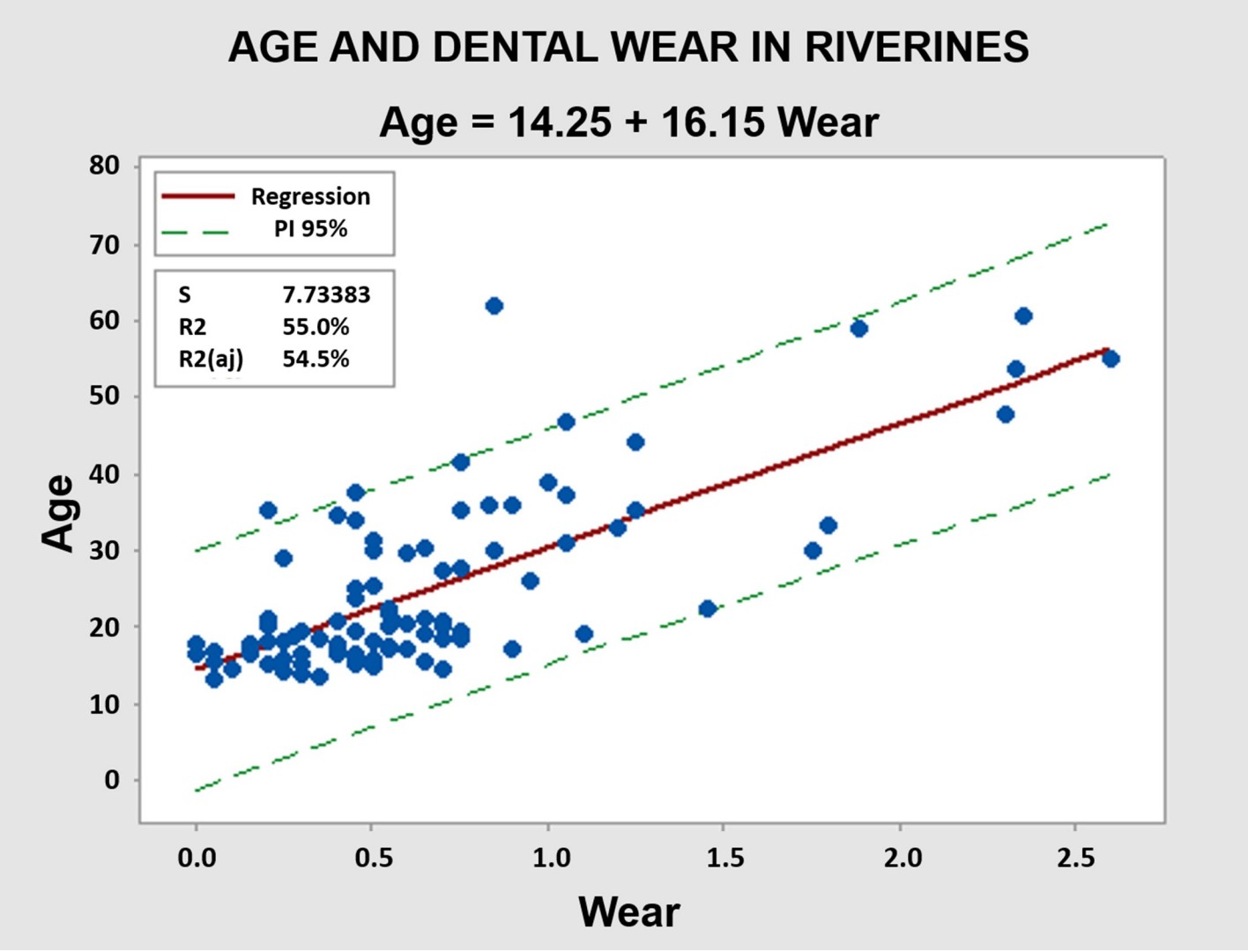

**Fig 2. Association between tooth wear and age for the riverine population.** Linear regression was statistically significant (p <0.001), explaining 55% of age variability in this group. *p<0.05.

age variation in these individuals. This value is intermediate to that obtained in semi-isolated indigenous populations and urban populations [11].

The amount of tooth wear present in a person depends on many factors [15, 26–28]. Therefore, quantifying dental wear to predict chronological age requires specific prediction models for each population to control these confounding factors which may vary between different cultures. Thus, extrapolation of our results to other populations requires caution.

There are several well accepted indices in the literature to measure tooth wear, and to compare different population groups through different indices can lead to wrong conclusions [15]. For example, in Chinese Malaysian and Indian populations, a high correlation was found between tooth wear and age [9, 10], similar to that seen in Xingu indigenous populations. Nevertheless, another measurement index was used, in addition to different selection criteria. Thereby, assume that this proximity in prediction capacity is due to similar cultural habits is incorrect. In this study, the association between dental wear and age was compared for each population using the same measurement index, examiners and eligibility criteria.

Our results show an association between tooth wear and chronological age in riverine people intermediate to that found in indigenous groups and urban population. The strongest association between the variables was observed in the indigenous groups mainly due to the fact that they present customs that cause greater tooth abrasion, such as their diet based on abrasive and unrefined foods [15, 29, 30]. A national survey on health and nutrition of indigenous people in Brazil showed that the main sources of food for groups living in the north are the cultivation and/or raising animals, in addition to hunting and fishing, which was reported less frequently in other regions [31]. In addition, indigenous people often use their teeth as "tools," serving as a third hand for holding, cutting and even breaking objects [24, 32]. In contrast, the people in the urban populations eat processed foods, which are less abrasive, and use their teeth primarily to eat [15, 33]. More pronounced dental wear in these people is often due to the presence of para-functional habits, dental erosion and, more rarely, to intrinsic factors such as gastroesophageal reflux and alcoholism [14, 15, 34]. This leads to greater variability of wear in relation to age, with individuals of different age groups presenting similar levels of wear [11]. This way, the association between age and tooth wear in this group is less.

In turn, riverine people have a diet and behavior similar to those of indigenous communities, such as the consumption of regional roots, cassava flour, local fruits and vegetables, fish or meat of wild animals [16, 34]. Cassava is obtained through slash and burn agriculture and is the most common crop planted both in the wetland and upland environments, mostly to be transformed into cassava flour. Most riverines families produce cassava flour only for their own consumption. In some cases, when families can work together, they may produce it in sufficient amounts to be sold in the nearby city [16, 35]. In general, fish is by far the main animal protein consumed, both in the rainy season and in the dry season, being present in approximately 60% of meals. Other sources of protein such as peccary (*Tayassu pecari*), tortoise (*Chelonoidis spp*.) and paca (*Cuniculus paca*) increase their participation in the diet during the dry season [35,36]. Meanwhile, the urban development of the region and government financial programs allowed to this population more access to urban centers and, consequently, increased the consumption of processed food beyond the acquisition of new habits [16, 18, 35]. Probably because of this, the association between tooth wear and chronological age observed in this group was less than that found in the indigenous and more than the urban population.

The process of cultural contact through which a person or group adopts values and practices from another culture, to a greater or lesser extent, is called acculturation. Ethnic minorities living with or in contact with multicultural societies face two key questions: maintain their ethnic identity or acquire behaviors from another culture [19–21]. As a result, there may be four situations: integration, assimilation, separation, or marginalization. Integration represents people who maintain their native culture but also adopt new culture habits. Assimilation refers to those who adopt the habits and behaviors of the new culture without maintaining their traditional customs. Separation occurs when individuals do not recognize or engage in new culture attitudes and behaviors and keep their original culture intact. And marginalization occurs when a group does not engage with another culture, but also fails to preserve its original [19–21, 37].

In the case of Amazonian riverine people, integration best defines their relationship with the urban population, since the cultural base is maintained, though, the eating habits of the other culture were adopted. Dietary acculturation is characterized when members of a minority group adopt eating patterns/food choices from a greater group. This phenomenon can cause a transition in the nutrition pattern of the riverines, which may reflect in weakening of the association between tooth wear and chronological age [38]. This reflected in the weakening of the association between tooth wear and chronological age. Thus, it is possible to suggest that

this relationship can indicate loss of cultural identity of traditional Amazonian populations since it is strong in semi-isolated communities and weakens with more contact with urban centers.

Thereby, dental wear can be used to estimate chronological age in Amazonian riverine residents, as a significant association was found between these variables. However, the reliability of this relationship decreases as a result of contact with the urban environment. Therefore, tooth wear, a strong evidence of the individual's eating habits in the past, may be an indicator of the acculturation process, and consequent cultural identity loss of traditional Amazonian populations.

## Supporting information

**S1 File. Approval of the Research Ethics Committee of the Health Sciences Institute of the Federal University of Pará.**
(DOCX)

## Acknowledgments

We thank Elma Vieira Takeuchi for her contribution to the conceptualization of the work.

## Author Contributions

**Conceptualization:** David Normando, Mayara Silva Barbosa, Cátia Quintão.

**Data curation:** David Normando, Mayara Silva Barbosa, Paulo Mecenas, Cátia Quintão.

**Formal analysis:** David Normando, Mayara Silva Barbosa, Paulo Mecenas, Cátia Quintão.

**Investigation:** David Normando, Mayara Silva Barbosa, Cátia Quintão.

**Methodology:** David Normando, Mayara Silva Barbosa, Cátia Quintão.

**Project administration:** David Normando, Mayara Silva Barbosa.

**Supervision:** David Normando, Cátia Quintão.

**Validation:** David Normando, Cátia Quintão.

**Visualization:** David Normando, Cátia Quintão.

**Writing – original draft:** Mayara Silva Barbosa, Paulo Mecenas.

**Writing – review & editing:** David Normando, Paulo Mecenas, Cátia Quintão.

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
