## [Decision Letter · Decision Letter 0]

6 Jan 2020

PONE-D-19-25583

Tooth wear as an indicator of acculturation process in remote Amazonian populations

PLOS ONE

Dear Mr. Mecenas,

Thank you for submitting your manuscript to PLOS ONE. After careful consideration, we feel that it has merit but does not fully meet PLOS ONE’s publication criteria as it currently stands. Therefore, we invite you to submit a revised version of the manuscript that addresses the points raised during the review process.

I now have several expert opinions on this work. Although all felt this work had merit all reviewers identified significant background logic, methodological, and contextual concerns with the manuscript as written. In a revision we request you address each of the reviewers major concerns.

We would appreciate receiving your revised manuscript by Feb 20 2020 11:59PM. To enhance the reproducibility of your results, we recommend that if applicable you deposit your laboratory protocols in protocols.io, where a protocol can be assigned its own identifier (DOI) such that it can be cited independently in the future. For instructions see: http://journals.plos.org/plosone/s/submission-guidelines#loc-laboratory-protocols

We look forward to receiving your revised manuscript.

Kind regards,

JJ Cray Jr., Ph.D.

Academic Editor

PLOS ONE

Journal Requirements:

2. PLOS ONE's publication criteria require that experiments, statistics, and other analyses are performed to a high technical standard; sample sizes are large enough to produce robust results; and methods are described in sufficient detail to allow another researcher to reproduce the experiment (http://journals.plos.org/plosone/s/criteria-for-publication#loc-3). If materials, methods, and protocols are well established, authors may cite articles where those protocols are described in detail, but the submission should include sufficient information to be understood independent of these references (https://journals.plos.org/plosone/s/submission-guidelines#loc-materials-and-methods). In this case, please ensure that you provide sufficient methodological detail even if they are available in your earlier publication at https://journals.plos.org/plosone/article?id=10.1371/journal.pone.0116138

3. We note that Figure 1 in your submission contain satellite image which may be copyrighted. All PLOS content is published under the Creative Commons Attribution License (CC BY 4.0), which means that the manuscript, images, and Supporting Information files will be freely available online, and any third party is permitted to access, download, copy, distribute, and use these materials in any way, even commercially, with proper attribution. For these reasons, we cannot publish previously copyrighted maps or satellite images created using proprietary data, such as Google software (Google Maps, Street View, and Earth). For more information, see our copyright guidelines: http://journals.plos.org/plosone/s/licenses-and-copyright.

a) You may seek permission from the original copyright holder of Figure 1 to publish the content specifically under the CC BY 4.0 license. 

Reviewers' comments:

Reviewer's Responses to Questions

**Comments to the Author**

1. Is the manuscript technically sound, and do the data support the conclusions?

Reviewer #1: Partly

Reviewer #2: Partly

Reviewer #3: Partly

2. Has the statistical analysis been performed appropriately and rigorously? 

Reviewer #1: No

Reviewer #2: I Don't Know

Reviewer #3: Yes

3. Have the authors made all data underlying the findings in their manuscript fully available?

Reviewer #1: Yes

Reviewer #2: Yes

Reviewer #3: Yes

4. Is the manuscript presented in an intelligible fashion and written in standard English?

Reviewer #1: Yes

Reviewer #2: Yes

Reviewer #3: Yes

5. Review Comments to the Author

Reviewer #1: This article is an analysis of the relationship between dental wear and age in an Amazonian Riverine population. The authors correlated the summed score of dental wear of the anterior teeth and premolars against personal age using a linear correlation. The resultant R squared value is intermediate between an urban population (very little relationship) and more isolated indigenous populations. The authors argue that tooth wear can be an indicator of the acculturation process, serving as a tool to assess the loss of cultural identity of traditional Amazonian populations.

There are a series of problems here - theoretical and methodological. Tooth wear is a product of the abrasiveness of food, patterns of occlusion, use of teeth as tools - as the authors point out. However, these are a relatively small part of any acculturation process. The authors use the four field model of acculturation: segregation etc etc. This model points out that there is no single trajectory in acculturation, that individual and group dimensions may vary (people may retain culturally significant values while participating in a broader economy for instance), and more recent work points out acculturation is a two way process between the groups involved. What the dental wear reflects is that as access to different types of food and technology changes so does the relationship between wear and age but this by itself is only part of acculturation and cannot used to assess the loss of cultural identity (as the authors argue in the last line of their article). Effectively the authors are asking a narrower question: does the relationship between age and dental wear alter as diet and technology change. Their data would suggest yes and this falls within the findings from a large series of other studies including extensive studies from the 1950s and 1960s at Yuendumu Central Australia.

However, in demonstrating this the authors need to consider the nature of their sample which includes people born between c2005 and 1957. In other words a population which has undergone extensive change - the younger people in their group have not experienced the same changes as the older individuals. The slope of the line is an artefact of the sample not proof that younger people will experience the same rate of dental wear as they age. In particular if the sample was divided by birth cohorts you will probably find different relationships since in figure 3 the slope for the very youngest individuals is flat. The authors need to look at the distribution of the residuals.

Furthermore their current sample includes older individuals than their comparative samples and in this instance it is the older individuals who seem to be pulling the slope so the R squared values are not strictly comparable given the structure of the sample. I recommend the authors think more carefully about the nature of the samples and the regressions.

The study is an interesting demonstration that the relationship between dental wear and age is population specific and reflects aspects of food availability and technology and that if that relationship is known for a specific population then it might be used as one way of assessing personal age of the unknown dead. However, using dental wear as a measure of acculturation is to ignore the work on acculturation and how it operates upon different dimensions.

Reviewer #2: This manuscript purports to investigate the correlation between tooth wear and chronological age in a riverine population of the Amazon. Valuable aspects of this research include its comparative aspect and the fact that urban and indigenous samples were studied by the same research team that studied the riverine group using the same methodology. This feature of the work enhances reliability and minimizes the possibility that results are impacted by differences in methodology or research protocol.

Some issues discussed in the manuscript would benefit from further consideration include the description of subsistence, use of teeth as tools, and statistical analysis of intra- and inter-observer variation in observing and recording data on tooth wear.

I would like to see a more detailed account of subsistence methods. In the abstract we are informed that the riverine groups rely on nature for subsistence, which is quite general. Later the text states that subsistence includes small-scale fishing and agriculture, plus hunted wild meat. It would be informative for the reader to have estimates of what percent of the diet comes from farming / agriculture; fishing; and hunting. Do the riverine groups forage of collect food?, what kinds? Or does all the vegetarian portion of the diet come from agriculture? Can the type of agriculture be more precisely characterized? Is the farming horticultural? swidden? Are field left fallow? for how long? Please clarify. In the same regard, how similar or variable is the mode of subsistence in different study groups? This is important in understanding inter-group variation in diet and its impact on the degree of occlusal tooth wear.

In terms of methodology, especially the assessment of degree of tooth wear, were all subjects examined by one researcher? If so, was a statistical analysis of intra-observer reliability or repeatability of degree of dental wear conducted? If more than one researcher participated in the observation / scoring of dental wear, was a statistical assessment of concordance or of discordance between observers conducted? This is an important way in which the rigor in dental data collection can be assessed and should be routinely included in such studies.

Less important issues that need attention include redundancy, accuracy of wording, and issues with the references.

Wording & phrases that need reconsideration:

in the abstract and elsewhere the authors say that riverine populations inhabit river borders, this is redundant and should be revised.

p. 3, line 61, for example: is doubly redundant. “..living along the river borders, live the riverine people…” living populations…live; and river borders …riverine. Some revision of the text is required here.

p. 3, line 49: the authors state that mineralization and tooth eruption are inaccurate methods for mature individuals. I contend this should be changed to read that timing of mineralization and tooth eruption cannot be used in estimating the chronological age of mature individuals

p. 3, line 66 (and elsewhere, p. 10, line 207, for example) use of teeth as tools is mentioned, glancing at the references does not provide further information on How teeth are used as tools?

Please provide brief examples. In fact one of the references for use of teeth as tools appears to focus on dental caries prevalence, not tooth wear.

In several places in the References section extraneous information is provided in addition to the journal name. This is unnecessary and should be deleted.

p. 14, line 319: the official journal of the Human Biology Council.

p. 15, line 327: a journal of the Association for Psychological Science.

p. 16, line 346: official publication of the American Association of Orthodontists, its

constituent societies, and the American Board of Orthodontics.

p. 16, line 363: Bericht uber die biologisch-anthropologische Literatur.

additionally multiple references to electronic sources, such as doi, ePub, PubMed, PubMed Central and others are given for the same reference - this seem unnecessary and redundant, a single source for each publication should be sufficient.

p. 16, line 348 (for example): Epub 2016/11/23. doi: 10.1016/j.ajodo.2016.03.033. PubMed PMID: 27871711.

and line 352 (for another example) Epub 2011/09/14. doi: 10.1016/j.jdent.2011.08.014. PubMed PMID: 21911033.

Reviewer #3: Interesting sociological and anthropological article. The authors are quite experts in the field and have already published about these topics in different journals. I just believe that the discussion is a little bit too short concerning the acculturation process. The authors have just cited one (main) book dated from 2003 and one article. May be it could be more developped and put in perspective with other studies (if possible).

6. PLOS authors have the option to publish the peer review history of their article (what does this mean?). If published, this will include your full peer review and any attached files.

Reviewer #1: No

Reviewer #2: No

Reviewer #3: No

---

## [Author Response · Author response to Decision Letter 0]

17 Feb 2020

Journal Requirements:

 “Please ensure that your manuscript meets PLOS ONE's style requirements, including those for file naming.

 Authors: We adjusted the manuscript in accordance with PLOS ONE’s style.

 “PLOS ONE's publication criteria require that experiments, statistics, and other analyses are performed to a high technical standard; sample sizes are large enough to produce robust results; and methods are described in sufficient detail to allow another researcher to reproduce the experiment (http://journals.plos.org/plosone/s/criteria-for-publication#loc-3). If materials, methods, and protocols are well established, authors may cite articles where those protocols are described in detail, but the submission should include sufficient information to be understood independent of these references (https://journals.plos.org/plosone/s/submission-guidelines#loc-materials-and-methods). In this case, please ensure that you provide sufficient methodological detail even if they are available in your earlier publication at https://journals.plos.org/plosone/article?id=10.1371/journal.pone.0116138.”

 Authors: Thank you for your remark. We adjusted the manuscript providing details about the methodology used.

 “We note that Figure 1 in your submission contain satellite image which may be copyrighted. All PLOS content is published under the Creative Commons Attribution License (CC BY 4.0), which means that the manuscript, images, and Supporting Information files will be freely available online, and any third party is permitted to access, download, copy, distribute, and use these materials in any way, even commercially, with proper attribution. For these reasons, we cannot publish previously copyrighted maps or satellite images created using proprietary data, such as Google software (Google Maps, Street View, and Earth). For more information, see our copyright guidelines: http://journals.plos.org/plosone/s/licenses-and-copyright.”

 Authors: Thank you for your remark. We have chosen to remove the Figure 1 from the manuscript.

Reviewers Requirements:

Reviewer #1:

“There are a series of problems here - theoretical and methodological. Tooth wear is a product of the abrasiveness of food, patterns of occlusion, use of teeth as tools - as the authors point out. However, these are a relatively small part of any acculturation process. The authors use the four field model of acculturation: segregation etc etc. This model points out that there is no single trajectory in acculturation, that individual and group dimensions may vary (people may retain culturally significant values while participating in a broader economy for instance), and more recent work points out acculturation is a two way process between the groups involved. What the dental wear reflects is that as access to different types of food and technology changes so does the relationship between wear and age but this by itself is only part of acculturation and cannot used to assess the loss of cultural identity (as the authors argue in the last line of their article).”

Authors: We appreciated your suggestion and we agree. However, our intention was not to support the use of dental wear to quantify the loss of cultural identity, our aim is to provoke that it can be used as quantitative indicator of acculturation. The sentence in the last line of the article did not really pass that thought. We modified the discussion and conclusion to make our intention clearer.

 “Effectively the authors are asking a narrower question: does the relationship between age and dental wear alter as diet and technology change.” 

Authors: The influence of changing dietary habits and access to technology in the relationship between age and dental wear helps us to support the idea of this study, however we absolutely agree that this is not the main issue in our work. Our hypothesis is that interaction between the riverside population and the urban population changed habits of the former, which reflected in changes in the pattern of tooth wear and consequently in its relationship with age. This may suggest that the acculturation process, but we cannot really evaluate this process in general, since the relationship between dental wear and acculturation is only part of it, as you mentioned.

“Their data would suggest yes and this falls within the findings from a large series of other studies including extensive studies from the 1950s and 1960s at Yuendumu Central Australia.

However, in demonstrating this the authors need to consider the nature of their sample which includes people born between c2005 and 1957. In other words a population which has undergone extensive change - the younger people in their group have not experienced the same changes as the older individuals. The slope of the line is an artefact of the sample not proof that younger people will experience the same rate of dental wear as they age. In particular if the sample was divided by birth cohorts you will probably find different relationships since in figure 3 the slope for the very youngest individuals is flat. The authors need to look at the distribution of the residuals.”

Authors: Thank you for your suggestion. Indeed, it would be ideal to have a larger number of older people in the sample, however, according to our exclusion criteria, the individual could not have more than 7 missing teeth, which tends to exclude older participants. Note that the slope of the line for young people is caused by the concentration of wear scores between 0 and 1, reflecting the absence of wear or wear only in tooth enamel, which is “normal” for this age group. If we divide into birth cohorts, we can have a more accurate view, but more limited and with a lower power for comparison with the samples already evaluated.

“Furthermore their current sample includes older individuals than their comparative samples and in this instance it is the older individuals who seem to be pulling the slope so the R squared values are not strictly comparable given the structure of the sample. I recommend the authors think more carefully about the nature of the samples and the regressions.”

Authors: Thank you for your remark. Mean age is not much higher compared to comparative samples. The difference is approximately only 2 years for the urban population and 3 for the average indigenous population. Older individuals tend to pull the slope since they are prone to more dental wear, this inherent to the characteristic which we are evaluating. Interestingly, this was not observed in the urban population, probably due to factors besides diet which can lead to tooth wear in young people. Furthermore, we are not comparing mean difference of tooth wear only. Tooth wear is adjusted by age in the regression model.

“The study is an interesting demonstration that the relationship between dental wear and age is population specific and reflects aspects of food availability and technology and that if that relationship is known for a specific population then it might be used as one way of assessing personal age of the unknown dead. However, using dental wear as a measure of acculturation is to ignore the work on acculturation and how it operates upon different dimensions.”

Authors: We appreciate all your suggestions. We modified the text to make it clear that dental wear can be an indicator of acculturation and not a tool for measuring this process.

Reviewer #2:

“I would like to see a more detailed account of subsistence methods. In the abstract we are informed that the riverine groups rely on nature for subsistence, which is quite general. Later the text states that subsistence includes small-scale fishing and agriculture, plus hunted wild meat. It would be informative for the reader to have estimates of what percent of the diet comes from farming / agriculture; fishing; and hunting. Do the riverine groups forage of collect food?, what kinds? Or does all the vegetarian portion of the diet come from agriculture? Can the type of agriculture be more precisely characterized? Is the farming horticultural? swidden? Are field left fallow? for how long? Please clarify. In the same regard, how similar or variable is the mode of subsistence in different study groups? This is important in understanding inter-group variation in diet and its impact on the degree of occlusal tooth wear.”

Authors: Thank you for your remark. We added more information to the text regarding the dietary and subsistence habits of riverines and indigenous people in general, but some variability according to the community studied is quite common. Very specific information about the type of agriculture practiced, such as fallow time, was not added due to the literature scarcity and we have not collected it systematically.

“In terms of methodology, especially the assessment of degree of tooth wear, were all subjects examined by one researcher? If so, was a statistical analysis of intra-observer reliability or repeatability of degree of dental wear conducted? If more than one researcher participated in the observation / scoring of dental wear, was a statistical assessment of concordance or of discordance between observers conducted? This is an important way in which the rigor in dental data collection can be assessed and should be routinely included in such studies.”

Authors: Thank you for your observation. There was prior calibration of one researcher who obtained the measurements and, afterwards, the intraclass correlation test revealed excellent reproducibility for the measurements. This was added to the manuscript.

“Less important issues that need attention include redundancy, accuracy of wording, and issues with the references.

Wording & phrases that need reconsideration: in the abstract and elsewhere the authors say that riverine populations inhabit river borders, this is redundant and should be revised.

p. 3, line 61, for example: is doubly redundant. “..living along the river borders, live the riverine people…” living populations…live; and river borders …riverine. Some revision of the text is required here.

p. 3, line 49: the authors state that mineralization and tooth eruption are inaccurate methods for mature individuals. I contend this should be changed to read that timing of mineralization and tooth eruption cannot be used in estimating the chronological age of mature individuals.”

Authors: We removed redundant phrases from the manuscript, improved the accuracy of wording and adjusted the references. Thank you.

“p. 3, line 66 (and elsewhere, p. 10, line 207, for example) use of teeth as tools is mentioned, glancing at the references does not provide further information on How teeth are used as tools?

Please provide brief examples. In fact one of the references for use of teeth as tools appears to focus on dental caries prevalence, not tooth wear.”

Authors: Thank you for your remark. Teeth can be used as tools to cut food or fishing lines, to hold artifacts while fishing or even to break things. However, it is difficult to find a reference for the occurrence of these habits, as there are no studies that describe them. Thus, we decided to remove this sentence from the text.

“In several places in the References section extraneous information is provided in addition to the journal name. This is unnecessary and should be deleted.

p. 14, line 319: the official journal of the Human Biology Council.

p. 15, line 327: a journal of the Association for Psychological Science.

p. 16, line 346: official publication of the American Association of Orthodontists, its constituent societies, and the American Board of Orthodontics.

p. 16, line 363: Bericht uber die biologisch-anthropologische Literatur.

Additionally, multiple references to electronic sources, such as doi, ePub, PubMed, PubMed Central and others are given for the same reference - this seem unnecessary and redundant, a single source for each publication should be sufficient.

p. 16, line 348 (for example): Epub 2016/11/23. doi: 10.1016/j.ajodo.2016.03.033. PubMed PMID: 27871711.

and line 352 (for another example) Epub 2011/09/14. doi: 10.1016/j.jdent.2011.08.014. PubMed PMID: 21911033.”

Authors: Thank you for your remarks. We remove all unnecessary information.

Reviewer #3: Interesting sociological and anthropological article. The authors are quite experts in the field and have already published about these topics in different journals. I just believe that the discussion is a little bit too short concerning the acculturation process. The authors have just cited one (main) book dated from 2003 and one article. May be it could be more developped and put in perspective with other studies (if possible).

Authors: Thank you for all your suggestions. We improved the discussion concerning the acculturation process and added more references to describe this process. However, it is difficult to put other studies in perspective due to the scarcity of works on the acculturation process in riverines communities or similar populations.

---

## [Decision Letter · Decision Letter 1]

10 Mar 2020

Tooth wear as an indicator of acculturation process in remote Amazonian populations

PONE-D-19-25583R1

Dear Dr. Mecenas,

We are pleased to inform you that your manuscript has been judged scientifically suitable for publication and will be formally accepted for publication once it complies with all outstanding technical requirements.

With kind regards,

JJ Cray Jr., Ph.D.

Academic Editor

PLOS ONE

Additional Editor Comments (optional):

Reviewers' comments:

Reviewer's Responses to Questions

**Comments to the Author**

1. If the authors have adequately addressed your comments raised in a previous round of review and you feel that this manuscript is now acceptable for publication, you may indicate that here to bypass the “Comments to the Author” section, enter your conflict of interest statement in the “Confidential to Editor” section, and submit your "Accept" recommendation.

Reviewer #3: (No Response)

2. Is the manuscript technically sound, and do the data support the conclusions?

Reviewer #3: Yes

3. Has the statistical analysis been performed appropriately and rigorously? 

Reviewer #3: Yes

4. Have the authors made all data underlying the findings in their manuscript fully available?

Reviewer #3: Yes

5. Is the manuscript presented in an intelligible fashion and written in standard English?

Reviewer #3: Yes

6. Review Comments to the Author

Reviewer #3: All the comments I made have been taken into consideration by the authors. I consider this paper acceptable for publication.

7. PLOS authors have the option to publish the peer review history of their article (what does this mean?). If published, this will include your full peer review and any attached files.

Reviewer #3: No

---

## [Editor Report · Acceptance letter]

20 Mar 2020

PONE-D-19-25583R1 

Tooth wear as an indicator of acculturation process in remote Amazonian populations 

Dear Dr. Mecenas:

I am pleased to inform you that your manuscript has been deemed suitable for publication in PLOS ONE. Congratulations! Your manuscript is now with our production department. 

With kind regards,

on behalf of

Dr. JJ Cray Jr. 

Academic Editor

PLOS ONE